# Validation of Spatiotemporal and Kinematic Measures in Functional Exercises Using a Minimal Modeling Inertial Sensor Methodology

**DOI:** 10.3390/s20164586

**Published:** 2020-08-15

**Authors:** Benjamin R. Hindle, Justin W.L. Keogh, Anna V. Lorimer

**Affiliations:** 1Faculty of Health Sciences and Medicine, Bond University, Gold Coast 4226, Australia; jkeogh@bond.edu.au (J.W.L.K.); alorimer@bond.edu.au (A.V.L.); 2Sports Performance Research Institute New Zealand (SPRINZ), AUT Millennium Institute, AUT University, Auckland 0632, New Zealand; 3Cluster for Health Improvement, Faculty of Science, Health, Education and Engineering, University of Sunshine Coast, Sunshine Coast 4556, Australia; 4Kasturba Medical College, Mangalore, Manipal Academy of Higher Education, Manipal, Karnataka 576104, India

**Keywords:** biomechanics, kinematics, spatiotemporal, gait, motion analysis, inertial sensors

## Abstract

This study proposes a minimal modeling magnetic, angular rate and gravity (MARG) methodology for assessing spatiotemporal and kinematic measures of functional fitness exercises. Thirteen healthy persons performed repetitions of the squat, box squat, sandbag pickup, shuffle-walk, and bear crawl. Sagittal plane hip, knee, and ankle range of motion (ROM) and stride length, stride time, and stance time measures were compared for the MARG method and an optical motion capture (OMC) system. The root mean square error (RMSE), mean absolute percentage error (MAPE), and Bland–Altman plots and limits of agreement were used to assess agreement between methods. Hip and knee ROM showed good to excellent agreement with the OMC system during the squat, box squat, and sandbag pickup (RMSE: 4.4–9.8°), while ankle ROM agreement ranged from good to unacceptable (RMSE: 2.7–7.2°). Unacceptable hip and knee ROM agreement was observed for the shuffle-walk and bear crawl (RMSE: 3.3–8.6°). The stride length, stride time, and stance time showed good to excellent agreement between methods (MAPE: (3.2 ± 2.8)%–(8.2 ± 7.9)%). Although the proposed MARG-based method is a valid means of assessing spatiotemporal and kinematic measures during various exercises, further development is required to assess the joint kinematics of small ROM, high velocity movements.

## 1. Introduction

Motion capture is a fundamental component of many modern biomechanical analyses. Common technologies used for human motion capture include optical, image/video processing and electromagnetic-based systems [1]. Although considered the gold standard of motion capture, optical motion capture (OMC) systems are expensive, typically limited to a laboratory environment, and suffer from marker occlusion, often resulting in loss of data [2]. Image/video processing systems suffer from similar marker occlusion problems, as well as parallax and perspective error [3]. Electromagnetic systems are limited to slow movements due to a low sampling frequency and are susceptible to large errors where ferromagnetic disturbances are present in the environment [1]. The limitations of current motion capture technology, particularly for field-based research, have prompted researchers to explore alternate technology for human motion capture.

Advancements in inertial measurement unit (IMU) and magnetic, angular rate and gravity (MARG) technologies has seen the development of affordable, compact, and powerful devices [4]. Inertial measurement units measure the tri-axial angular rate and linear acceleration, while MARG devices also measure the tri-axial magnetic field strength. By attaching IMU/MARG devices to individual body segments and performing specialized processing of the output data, the position and orientation of each segment and the resultant kinematics of the body can be estimated [5]. High sampling rates, an affordable equipment cost, and the ability to stream data live or collect data directly on the device for future download make IMU/MARG technology an attractive alternative to traditional motion capture systems. Researchers have used both proprietary and researcher-developed IMU/MARG systems to measure human movement for a range of applications, including sporting [6,7,8,9,10,11], clinical [12,13,14,15], and ergonomic [16,17,18,19] applications. Literature investigating the validity of IMU/MARG motion capture for the assessment of human kinematics suggests that the accuracy of IMU/MARG motion capture is dependent on the task complexity, movement speed, sensor placement, specific kinematic parameter being analysed, and processing methodology used [20,21]. Processing methods described in previous validation studies of researcher-developed systems, particularly in the areas of sensor fusion and sensor to segment alignment, provide valuable information for the development of IMU/MARG motion capture technology.

In its most simplistic form, integration of the angular rate data of an IMU/MARG device provides an orientation estimation of the device with respect to its original orientation in a local coordinate frame [22]. Integration of the inherent bias within the angular rate data results in cumulative drift error over time [23]. The acceleration due to gravity measured by the accelerometer may be used to assist in correcting the attitude (inclination) component of this drift; however, the signal becomes corrupt when the device is in a non-quasi-static state [22]. Similarly, the magnetometer data provides a heading (horizontal direction) orientation and can be used to assist in correcting the heading component of the drift. However, this heading estimation is often corrupted by magnetic disturbances within the environment [23].

Sensor fusion leverages the most reliable components of accelerometer, gyroscope, and magnetometer orientation observations at each time point to provide an orientation estimation of the device in a local or global reference frame [24]. While proprietary systems use their own sensor fusion algorithms, the most common methods of sensor fusion incorporate versions of the complementary filter [12,25,26] and Kalman filter [27]. Previous literature suggests minimal differences in the orientation estimation accuracy between such sensor fusion methods [22,28,29]. The ability to further tune the Kalman filter using various noise and disturbance parameters is suggested to give Kalman filter-based approaches a slight accuracy advantage over complementary filter approaches, albeit at the expense of the computational load [22].

Once the orientation of the IMU/MARG device has been established, the coordinate system of the device must be aligned with the coordinate system of the segment to which it is attached. This process is known as sensor to segment alignment. Sensor to segment alignment methods described in previous validation studies of researcher-developed systems can be categorized as manual alignment with or without the use of specialized alignment devices [7,30]; static pose estimation [13,31]; functional calibration [32,33,34,35]; and most recently, deep learning [36]. Although the former three alignment methods have been shown to have a minimal effect on the overall agreement between OMC and IMU/MARG measures [37], the practicality of such sensor to segment alignment methods should be considered.

The manual alignment method (also commonly referred to as the technical anatomical alignment method) requires the precise alignment of the local coordinate system of the IMU/MARG device with the anatomical coordinate system of each segment. The manual alignment method is the least computationally expensive method [37]; however, it comes at the cost of requiring additional specialized calibration equipment or highly skilled persons to identify anatomical landmarks and place sensors according to these landmarks [7,30].

Static pose calibration methods remove some of the reliance on the precise alignment of each IMU/MARG device coordinate system with the respective segment coordinate system by allowing the arbitrary placement of all but one device [13,31]. Mathematical transformations are used to transform a known local sensor coordinate system into a known segment coordinate system via a global coordinate system. This method appears to be a common compromise between computationally simplistic manual alignment and more computationally expensive approaches.

Functional calibration techniques require the client to perform specific movements with the IMU/MARG devices arbitrarily positioned on each segment [32,33,34,35]. Numerical methods are then used to determine the segment or joint coordinate systems from the data collected during the calibration movements. While the functional calibration method allows the arbitrary positioning of all IMU/MARG devices, the computational cost in establishing segment/joint coordinate systems is generally greater than the manual alignment and static pose method [37]. Additionally, certain conditions may prevent some clients from performing the calibration movements [33].

Most recently, deep learning has been used to achieve sensor to segment alignment [36]. This state-of-the-art approach relies on a quantity of previously collected real or simulation motion data to train a model to identify the orientation of an arbitrarily positioned sensor and automatically align it with the segment coordinate system. Although this method is relatively new and has seen limited development, initial research suggests that the method may be computationally expensive and that it requires large sets of existing data for accurate model training [36].

As there is currently no standardized methodology for IMU/MARG motion capture for all applications, it is necessary to learn from the previous literature and validate any novel or application-specific IMU/MARG motion capture methodology. To the best of the authors’ knowledge, no previous literature has validated the use of MARG-based motion capture during functional fitness exercises [20,38], where highly dynamic movements result in large ranges of motion across multiple joints [39].

The aim of this study was to assess the validity of a minimal modeling MARG motion capture methodology (from here on referred to as the MARG method) for the estimation of spatiotemporal (stride length, stride time, and stance time) and kinematic (sagittal plane hip, knee, and ankle joint range of motion (ROM)) parameters when compared to those obtained using an OMC system during various functional fitness exercises. The MARG method uses a minimal modeling approach, which includes the alignment of the sensor to the segment, processing, and anatomical modeling assumptions.

## 2. Materials and Methods

### 2.1. Participants

Thirteen participants, including 10 males (27.6 ± 10.8 y, 82.6 ± 13.5 kg, 181.4 ± 6.2 cm) and three females (31.1 ± 9.6 y, 61.2 ± 5.0 kg, 162.4 ± 5.1 cm), with a broad range of anthropometric characteristics, were recruited for this study to account for body type differences within the fitness population. All participants were required to have undertaken some form of resistance or cardiovascular training of a minimum of twice per week for at least six months prior to testing and be free from any injury at the time of testing. Participants meeting the defined criteria provided written informed consent prior to commencing testing. The study was conducted in accordance with the Declaration of Helsinki and ethical approval was granted for all procedures used throughout the study by the Bond University Human Research Ethics Committee (BH00070).

### 2.2. Experimental Protocol

Analyzed movements were selected based on their transferability to a range of exercise-related movement patterns [39,40] and their ability to be performed in a laboratory environment (Figure 1). The following subsection provides a description of these five movements.

#### 2.2.1. Squat

Each participant performed three sets of five squat repetitions. Participants were instructed to cross their arms over their chest and perform the squats to a maximum comfortable depth at a self-selected cadence.

#### 2.2.2. Box Squat

Each participant performed three sets of five box squat repetitions. Participants were instructed to cross their arms over their chest and perform the squats to the depth of a wooden box with the following dimensions: height: 500 mm × depth: 300 mm × width: 400 mm.

#### 2.2.3. Sandbag Pickup

Each participant performed three sets of three sandbag pickup repetitions (sandbag mass: 10 kg, diameter: ~400 mm, length: ~400 mm). Participants were instructed to adopt a hybrid stoop and squat lifting technique whereby the participant would initialize the lift with relatively straight legs and a curved upper spine, before positioning the sandbag in their lap and standing using a technique similar to the stone lift from the sport of strongman.

#### 2.2.4. Shuffle Walk

Each participant performed three sets of four to six strides of a modified gait pattern across the test volume, simulating the technique they may use if they were carrying a heavy object. Participants were instructed to vary their stride rate, stride length, and stride width throughout and between sets.

#### 2.2.5. Bear Crawl

Each participant performed three sets of bear crawls across the test volume. Participants were instructed to assume a four-point stance position before performing two to three strides across the test volume.

### 2.3. OMC Marker Placement and Processing

A six-camera Bonita Vicon 3D OMC system (Vicon Motion Systems Ltd., Oxford, UK), sampling at 100 Hz, was used as the reference for joint ROM and spatiotemporal estimations [41]. The capture volume was approximately 3 m × 2 m × 2 m. Fifteen 14 mm reflective markers were attached to the landmarks reported in Figure 2. Clusters of four reflective markers were attached to the lateral shank and thigh of the participant. Joint angles were estimated via inverse kinematics using Visual3D software (Visual3D, C-motion, Inc.; Rockville, MD, USA) [42].

### 2.4. MARG Placement and Processing

Four MARG sensors (ImeasureU, Vicon Motion Systems Ltd., Oxford, UK) were fixed on a single side of the participant’s body (Figure 2 and Table 1). The location of each MARG sensor was selected for repetitive identification by untrained persons in the field and to minimize the effect of soft tissue artefacts [43]. Each sensor consisted of a triaxial accelerometer (±16 g), triaxial gyroscope (±2000°/s), and triaxial magnetometer (±4900 µT) with an on-board sampling rate of 1125 Hz (accelerometer and gyroscope) and 112.5 Hz (magnetometer). The Capture.U app (software version 1.1.843, Vicon Motion Systems Ltd., Oxford, UK), installed on an iPad Air 2 (iOS 13.3.1, Apple Inc., CA, USA), was used to initialize and synchronize MARG device data recording. Raw MARG data (see Appendix A) were processed using distinct methods for kinematic and spatiotemporal measures.

#### 2.4.1. Kinematic Measures

A modified method for determining joint angle kinematics based on Beravs, Reberšek, Novak, Podobnik, and Munih [31] was developed using a custom Matlab script (The Mathworks Inc., Natick, MA, USA) (Figure 3). The following pre-processing and sensor to segment alignment methods were used.

MARG data pre-processing: Two different methods for preparing the raw MARG data were used in order to determine the most appropriate method for the selected movement patterns. These will be referred to as the default (DEF) method and the tuned and filtered (TAF) method (Figure 4).

For both the DEF and TAF method, the effects of soft and hard iron magnetic disturbances on the raw magnetic field data were reduced by performing a calibration procedure [44]. For the TAF method, gyroscope data were passed through a sixth-order low-pass Butterworth filter with a cut-off frequency of 60 Hz. Filter parameters were established from a frequency analysis of data collected in pilot testing. Acceleration data remained unfiltered in both DEF and TAF methods, based on pilot testing results.

Acceleration, angular rate data (raw for DEF, filtered for TAF), and magnetic field data (calibrated for hard and soft iron effects) were passed into an attitude heading reference system (AHRS) fusion filter to estimate the orientation of each MARG device in the global reference frame (Sensor Fusion and Tracking Toolbox Release 2019a, The Mathworks Inc., Natick, MA, USA). The AHRS filter used a 9-axis indirect Kalman filter to model the error process of the system. The filter allowed initial device and tuning properties to be set for a given movement and environment. 

In the TAF method, device tuning properties and biases were established using a combination of a static dataset collected over a four-hour period, information from the device datasheet, and pilot testing data of each exercise. These properties included the following: Variance of accelerometer ((m/s^2^)^2^) and gyroscope ((rad/s)^2^) noise; variance of magnetometer disturbance noise (µT^2^); gyroscope offset drift ((rad/s)^2^); a compensation factor for linear acceleration drift [0, 1]; and the expected magnetic field strength due to the geographic location (Table 2). In the DEF method, all filter properties remained as the default properties set by Matlab and the Kalman filter were left to correct for these errors (see MEMS Industry Group [45] for further details). 

From the AHRS filter, a quaternion representation of each device in the global frame was established. Quaternion and direction cosine matrix (DCM) representations were used throughout processing to avoid singularities (gimbal lock) inherent when using a common Euler representation [26].

The orientation of the MARG sensor positioned on the foot was such that the *x*-axis of the MARG sensor pointed in the anterior/posterior direction of the segment qMARGfoGF. The cross product of the known foot segment anterior/posterior facing *x*-axis component of the DCM, and the vertical *z*-axis component of the DCM [0, 0, 1], allowed the *y*-axis component perpendicular to the two known axes to be found. From the orientation of the foot segment in the global frame, the orientation of all segments in the global frame could be assumed to be aligned as qseg.oGF and defined as per Figure 5.

Using the Hamilton product of the known initial orientations as described using quaternions, the transformation $q_{seg.t}$ of each MARG sensor’s initial orientation in the global frame to the initial segment orientation in the global frame could be determined using Equation (1), where * denotes the quaternion conjugate.
(1)qseg.t=qseg.o* GF⊗qMARGseg.oGF

Segment orientation at each time instance qseg.kGF could then be determined by taking the Hamilton product of the quaternion representation of the transformation of each MARG sensor to segment orientation and the orientation of the MARG sensor in the global frame at time instant k using Equation (2).
(2)qseg.k GF=qMARGseg.kGF⊗qseg.t*

Joint angles were calculated as the difference in orientation between a proximal qseg1kGF and distal segment qseg2kGF at each time instant, as described using quaternions (Equation (3)). A visual representation of the joint angle (difference in the quaternion orientation) could then be obtained using an Euler angle representation.
(3)qj=qseg1k*GF⊗qseg2kGF

#### 2.4.2. Spatiotemporal Measures

The stride and stance time were estimated using a custom Matlab script, from initial contact (IC) and final contact (FC) points identified from acceleration data using the methods of Jasiewicz et al. [46]. Stride length estimation was achieved using a zero velocity update (ZUPT) methodology [47]. The initial orientation estimation of the pelvis sensor was used to determine the foot segment coordinate system and direction of travel using the sensor to segment alignment methodology described above. The acceleration at the heel (minus acceleration due to gravity) was integrated using a trapezoidal approximation to give the velocity of the foot. The drift resulting from the integration of the motional acceleration was corrected by means of a ZUPT. Where a stance phase (and thus known instance of zero velocity) was detected, a Kalman filter was used to reduce the drift caused when integrating by approximating the error in the system. After the ZUPT correction, the stride length could be estimated as the distance travelled between consecutive stance phases.

### 2.5. Data Analysis and Statistical Methods

Data were first assessed for normality by visual inspection and a Shapiro Wilks test. The mean absolute percentage error (MAPE) and root mean squared error (RMSE) were calculated for each spatiotemporal and kinematic measure. A classification system was used to assess MAPE values [48], where MAPE ≤ 5% = excellent agreement, 5% < MAPE ≤ 10% = good agreement, 10% < MAPE ≤ 15% = acceptable agreement, and MAPE > 15% = unacceptable agreement. To provide greater insight into the agreement of joint angle estimations throughout the range of motion of each repetition, a measure of the percentage of time the MARG method error was within ±10% of the ROM of the OMC system was calculated (E_10%_). An acceptable error threshold of ±10% for the E_10%_ calculation was selected to show a clinical difference in means [49]. For time-series comparative measures, MARG joint angle approximations were resampled to 100 Hz and synchronized manually based on the point of maximum flexion throughout a repetition.

Bland–Altman upper and lower 95% limits of agreement (LoA) were used to assess agreement between methods [50,51]. The LoA were set to 1.96 times the upper and lower standard deviation of the difference between the OMC and MARG method. Where normality was not met, a log transformation was performed prior to undertaking the Bland–Altman analysis. Paired t-tests were conducted between TAF and DEF methods. A Wilcoxon signed-rank test was performed where data were not normally distributed. All statistical analyses were performed in R version 3.6.1 (R Development Core Team, Vienna, Austria), with statistical significance accepted at *p* < 0.05.

## 3. Results

### 3.1. Kinematic Measures

Hip, knee, and ankle joint ROM were compared for IMU and OMC during 195 squat, 195 box squat, and 117 sandbag pickup repetitions, while 193 hip and 195 knee, and 115 hip and 113 knee ROMs were compared for the modified gait and bear crawl, respectively. Marker dropout in the OMC prevented a comparison of hip and knee joints during three crawl strides and two modified gait strides. 

Hip and knee joint angle estimation using both the DEF and TAF method showed good to excellent agreement with the OMC system when performing repetitions of the squat, box squat, and sandbag pickup (Table 3). The root mean square error and MAPE of hip and knee ROM were less for the box squat than the squat when using the TAF MARG method. Bland–Altman plots indicate an underestimation in knee ROM for the squat and sandbag pickup when using the DEF method (Figure 6). The underestimation of knee ROM by the MARG method during the squat and sandbag pickup may reflect the large ROM (squat: 121.2 ± 9.5°; sandbag: 126.8 ± 7.2°) compared to the other three exercises. Although there were only three female participants out of the total sample of 13, when comparing data obtained from male and female participants (Figure 6), the underestimation in knee ROM during the squat, box squat, and sandbag pickup appeared to be larger in the female group than the combined or male group (DEF method), with such results also being apparent for the TAF method. Where no consistent bias was observed for the combined or male group, a slight overestimation in hip ROM by the MARG method in female participants (both DEF and TAF) may be observed during the squat, box squat, and sandbag pickup. Inconsistencies in the agreement between methods (combined group) were observed for both DEF and TAF methods through the relatively wide Bland–Altman LoA (Figure 6).

Ankle joint angle estimations generally showed good agreement with the OMC system when using the DEF method for the squat, box squat, and sandbag pickup (Table 3). When using the TAF method, acceptable (sandbag pickup) to unacceptable (squat and box squat) errors were observed. Bland–Altman plots indicate a slight (DEF) to moderate (TAF) overestimation bias in the MARG method for ankle joint ROM during the squat, box squat, and sandbag pickup exercises for the combined group (Figure 6). This overestimation (both DEF and TAF) appeared to be slightly smaller in female participants when compared to their male counterparts. Ankle ROM Bland–Altman LoA for the combined group were smallest for the box squat when compared to the squat and sandbag pickup.

In contrast to the squat, box squat, and sandbag pickup, unacceptable agreement at both the hip and knee joint was observed for the shuffle-walk and bear crawl, with the TAF method achieving slightly greater agreeance during the shuffle walk than the DEF method. Preliminary results indicated that a meaningful E_10%_ analysis of the hip and knee during the shuffle-walk and bear crawl could not be performed, with values ranging from 60.1% ± 23.9% to 78.4% ± 21.2%. This was in part due to the high noise to ROM ratio and slight phase duration discrepancy between the OMC and MARG method, as can be seen in the exemplar data provided in Figure 7. No consistent bias was observed for hip and knee ROM in the shuffle-walk and bear crawl (Figure 8), with wide LoA in both TAF and DEF methods further demonstrating the inconsistencies in measurements between the OMC and MARG method for hip and knee ROM (Table 4).

To an even greater extent than at the hip and knee, preliminary analysis of ankle joint ROM during the shuffle-walk and bear crawl resulted in a high noise to ROM ratio and unacceptably large MAPE. As such, it was determined that a meaningful comparison could not be performed and was omitted (Figure 7e,f).

### 3.2. Spatiotemporal Measures

The stride length, stride time, and stance times were compared for 192, 178, and 178 instances of the shuffle-walk, respectively, and 116, 83, and 83 instances of the bear crawl, respectively (Table 5). The stride length, stride time, and stance time MAPE showed good to excellent agreement with the OMC system (Table 6). Bland–Altman plots indicated a slight overestimation of the stride and stance time by the MARG method during both the shuffle-walk and bear crawl, and an underestimation of the stride length by the MARG method during the shuffle-walk (Figure 9 and Table 6).

## 4. Discussion

The aim of this study was to assess the validity of a minimal modeling MARG motion capture method for spatiotemporal and kinematic measures during repetitions of various functional fitness exercises. The MARG method used minimal modeling assumptions, in that simple, sensor to segment alignment, data processing (through the DEF method), and anatomical modeling assumptions were used. To the best of the authors’ knowledge, the exercises selected in the current study covered a wider range of sagittal plane ROM than previous literature [5,20,38,52]. 

The RMSE in hip, knee, and ankle ROM during the squat, box squat, and sandbag pickup were similar to those of previous research during squat, single leg squat, and counter movement jump exercises (hip: 4.9–8.3°; knee: 2.4–3.1°; ankle: 2.5–5.3°) [38]. While the knee ROM RMSE may be slightly greater in the current study than those of Teufl, Miezal, Taetz, Fröhlich, and Bleser [38], a MAPE of less than 10% was still considered to be a good level of agreement. Slightly greater agreeance was seen in joint ROM using the TAF method than the DEF method for the hip and knee; however, both methods were acceptable. The DEF method showed greater agreeance in all analysed exercises for ankle ROM and is suggested in preference to the TAF method for ankle joint measures. 

The shuffle-walk and bear crawl demonstrated small hip and knee joint ROM (12.1° ± 3.3–40.0° ± 20.4°) and agreement between the OMC system and both TAF and DEF MARG methods varied. Similar hip and knee RMSE during over-ground walking (hip: 6.1°, knee: 6.8°) have been found in previous studies [53]. The relatively large (>10%) MAPE found during the shuffle-walk and bear crawl movements in the current study suggest neither MARG method (DEF or TAF) may be acceptable for measuring the relatively moderate hip or knee ROM during the shuffle-walk or bear crawl. The high noise to ROM measurements observed in hip and knee ROM during the modified gait patterns (example seen in Figure 7d) made the manual alignment of OMC and MARG time-series plots based on peak values ambiguous. Furthermore, phase discrepancies were observed in these data (Figure 7a,b), which may be the result of the resampling of MARG joint angle estimations to 100 Hz for comparison with OMC. In exercises such as the shuffle walk where the stride duration is small (0.846 ± 0.219 s) relative to the sample rate of the OMC system (100 Hz), the modeling of few data points may result in the loss of fidelity in the joint angle approximation. As the MARG method is initially sampled and modeled at 1125 Hz, and then resampled to 100 Hz for comparison with the OMC, the loss of fidelity in the joint angle approximation may be less than the OMC approximation. The large differences observed in the timeseries curve analysis (in particular Figure 7e) may be a combined result of the inherent noise in the MARG method joint angle approximation and the loss of fidelity in the OMC joint angle approximation for short-duration activities, such as a stride in the shuffle walk and bear crawl. The ambiguity caused by both noise and phase duration discrepancy led to the inability to confidently report E_10%_ values for the hip and knee and as such, such data were omitted. It was concluded that the recommendation based on other error metric calculations, that neither MARG method (DEF or TAF) may be acceptable for measuring hip/knee ROM during the shuffle-walk or bear crawl, would not change upon the calculation of E_10%_ for all participants. 

Preliminary ankle ROM data of the shuffle-walk and bear crawl demonstrated an even greater noise to ROM ratio (Figure 7e,f) than at the hip and knee. Ambiguity caused by this large noise to ROM ratio lead to the inability to confidently report error metrics. Wells, Alderson, Camomilla, Donnelly, Elliott, and Cereatti [7] observed greater differences in OMC- and MARG-based joint angle estimations during higher velocity upper-limb sporting movements when compared to lower velocity movements. As the MARG devices used to measure ankle ROM are positioned closer to the extremity of the lower limb than those used to estimate hip and knee ROM, higher velocities and larger disagreement between the OMC and MARG joint angle estimation than at the hip and knee may be expected. Based on the preliminary data, error metrics of the hip and knee, and predicted greater error metrics at the ankle, it was concluded that neither MARG method may be suitable for ankle ROM assessment during the shuffle-walk and bear crawl where a small ROM and greater movement velocity are expected. 

Whilst previous literature has focused on comparing OMC relative angles of markers placed on or around MARG sensors to relative angles estimated from MARG, the current study compared biomechanically-modeled joint estimations derived from an OMC system to relative angles estimated from MARG measures. The relative angles measured using the MARG method assume that the anterior/posterior axis of the foot sensor and the anterior/posterior axes of all limbs are aligned during the calibration pose. Any error in the initial alignment will be apparent in the mathematical transformation of each individual segment sensor coordinate system to the respective segment coordinate system, with the error compounding where adjoining segments are misaligned. Brice et al. [54] demonstrated less agreement between OMC biomechanically-modeled joint angles and un-modeled MARG relative angles than OMC un-modeled relative angles and MARG measured relative angles. This leads to the suggestion that some of the differences in joint angle ROM estimations found in the current study may be due to the differences in modeling assumptions used in each of the OMC and MARG methods and the compounding error occurring throughout the alignment and mathematical transformation process. 

With the exception of the stride length, the errors in spatiotemporal measures during the shuffle-walk and bear crawl in the current study were greater than those observed using a similar methodology during over-ground walking [55]. The stride length, stride time, and stance time RMSE observed by Teufl, Lorenz, Miezal, Taetz, Fröhlich, and Bleser [55] during over-ground walking were 0.04 m, 0.01 s, and 0.02 s, respectively, with similar RMSE having been observed in treadmill running [56]. The larger disagreement in temporal parameters between the OMC and MARG method in the current study may partially be due to the difficulty in identifying the instance of IC and FC during the modified gait patterns, which resulted in reduced IC and reduced changes in heel acceleration during the initial swing than would be seen in a normal gait with longer strides [57]. In the modified gait patterns, identifying FC from a MARG sensor mounted on the lateral side of the heel, where the toe is the last true contact point with the ground, may lead to inaccuracies in identifying the FC instance. 

While this study addressed a number of gaps within the literature, a number of limitations of the current study should be noted. Data were only collected from a single side of the body, in a limited laboratory space and assessed only for sagittal plane flexion/extension ROM. Although a magnetic calibration was conducted for each testing session, it is expected that due to ferromagnetic disturbances present in the laboratory environment, the accuracy of the MARG method may have still been compromised. The reference OMC and MARG method use different physical measurements to derive joint angle estimations, with each method having associated noise. Measurement noise combined with different modeling assumptions would result in distinctly different noise properties and therefore signal patterns. The ability to compare estimations of small ROM between systems where the noise to signal ratio is high may be a major limitation when validating MARG against OMC methods [20,38].

## 5. Future Work

To further develop the proposed MARG method into an accurate means of measuring human kinematics during high velocity, small ROM movements, such as the shuffle walk and bear crawl, a number of areas of potential development are suggested. Further refinement of the Kalman filter tuning parameters, specialized for a given exercise (variance of accelerometer/gyroscope noise, and linear acceleration compensation factor) and the environment (magnetometer disturbance noise) may be needed to improve joint ROM estimations where high signal to noise ratios are observed [22]. These parameters may be established through further data collection and testing. Where previous literature has achieved segment coordinate system to sensor alignment using specialized equipment [7] or complicated movement-based algorithms [32,33,34,36], a possible middle-ground between the complexities of previous literature and the minimal methods used in the current study may be achieved. Although not a direct development of the MARG method, collecting data at a sampling rate common to both MARG and OMC equipment will likely result in greater agreeance between methods and provide a closer measure to the true validity of the MARG method. Future work should also look at assessing the validity of the MARG method for bilateral, multi-planar motion and assess its inter-day and assessor reliability.

## 6. Conclusions

The proposed minimal modeling MARG-based method is a valid means of assessing spatiotemporal and kinematic measures of persons performing various functional fitness exercises. It is suggested that care should be taken when selecting tuning and filtering parameters when using the MARG method for specific exercises. Although a high noise to joint ROM measurement ratio may be an inherent issue when assessing the validity of human motion analysis methods during some exercises, further development of the MARG method may result in a valid means of measuring small joint ROM during fast movements.

## Figures and Tables

**Figure 1 sensors-20-04586-f001:**
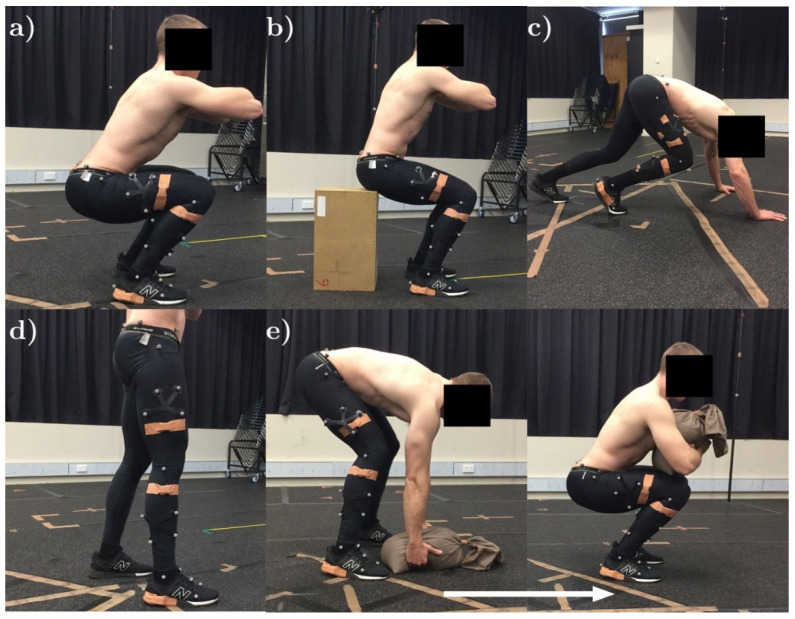
Functional fitness exercises: (**a**) Squat; (**b**) box squat; (**c**) bear crawl; (**d**) shuffle walk; and (**e**) sandbag pickup.

**Figure 2 sensors-20-04586-f002:**
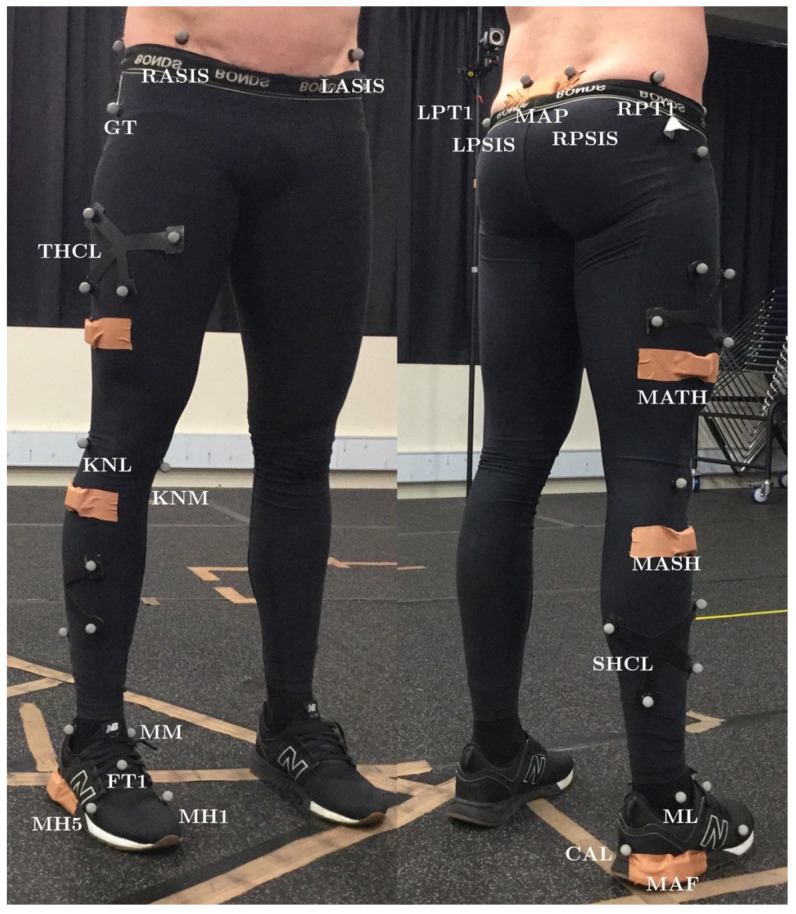
Optical motion capture (OMC) and magnetic, angular rate and gravity (MARG) sensor placement: CAL, calcaneus; FT1, foot tracking marker one; GT, greater trochanter; KNL, knee lateral; KNM, knee medial; LASIS, left anterior superior iliac spine; LPSIS, left posterior superior iliac spine; LPT1, left pelvis tracking marker one; MAF, foot MARG sensor; MAP, pelvis MARG sensor; MASH, shank MARG sensor; MATH, thigh MARG sensor; MH1, first metatarsal head; MH5, fifth metatarsal head; ML, lateral malleolus; MM, medial malleolus; RASIS, right anterior superior iliac spine; RPSIS, right posterior superior iliac spine; RPT1, right pelvis tracking marker one; SHCL, shank cluster; THCL, thigh cluster.

**Figure 3 sensors-20-04586-f003:**
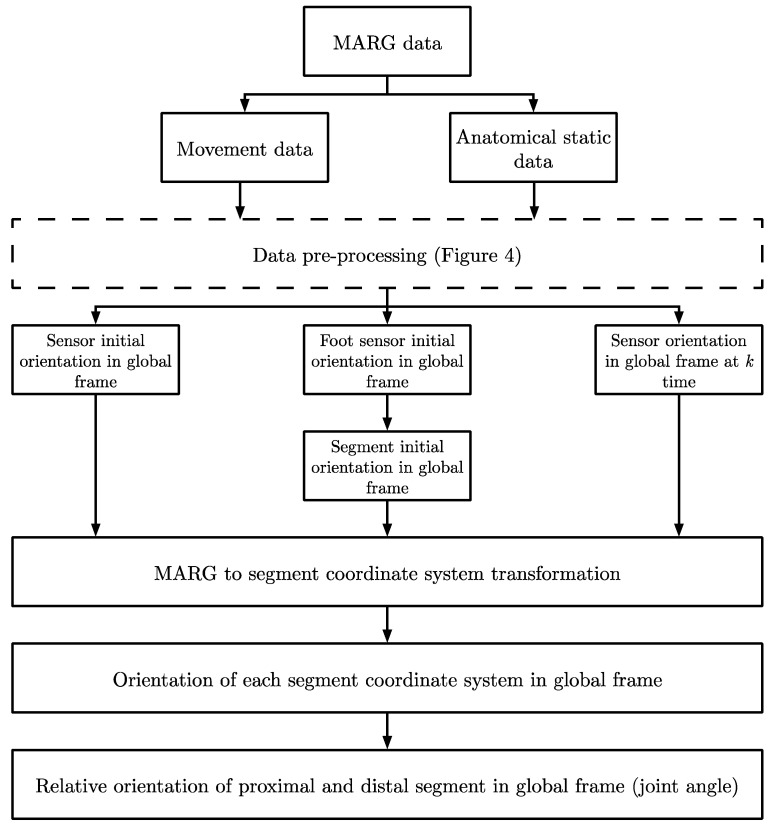
Joint angle estimation methodology overview.

**Figure 4 sensors-20-04586-f004:**
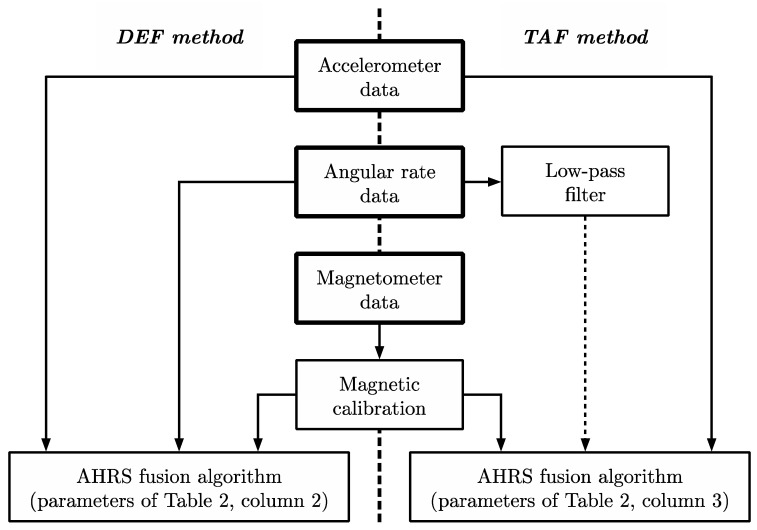
Data pre-processing default (DEF) and tuned and filtered (TAF) methods.

**Figure 5 sensors-20-04586-f005:**
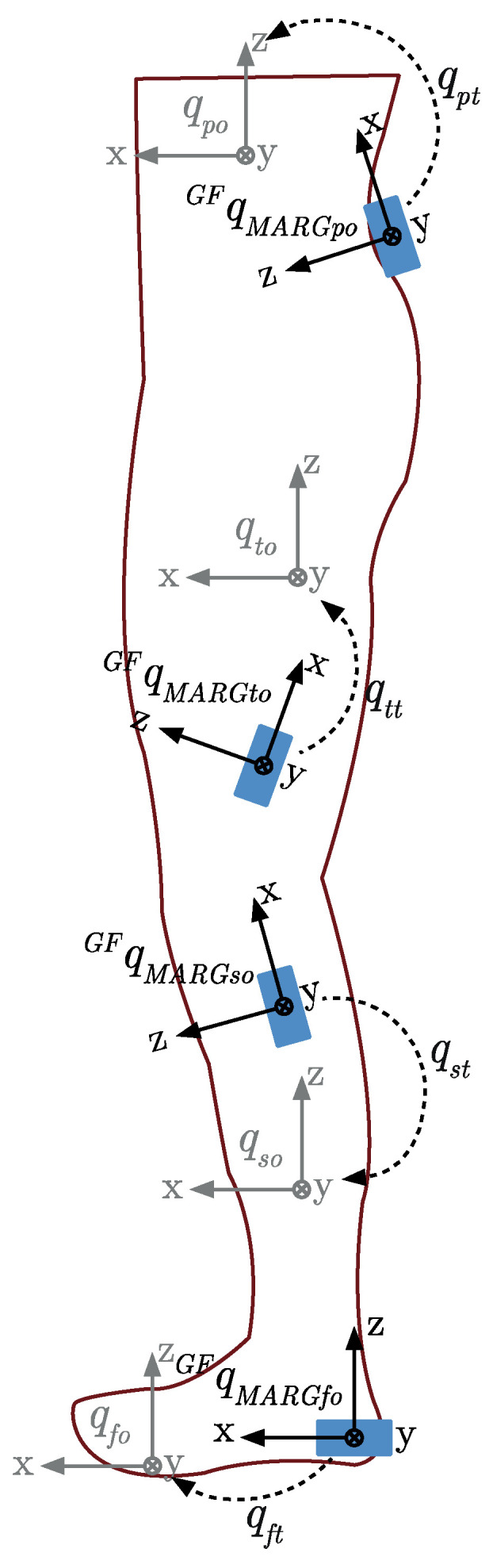
MARG sensor orientation transformations.

**Figure 6 sensors-20-04586-f006:**
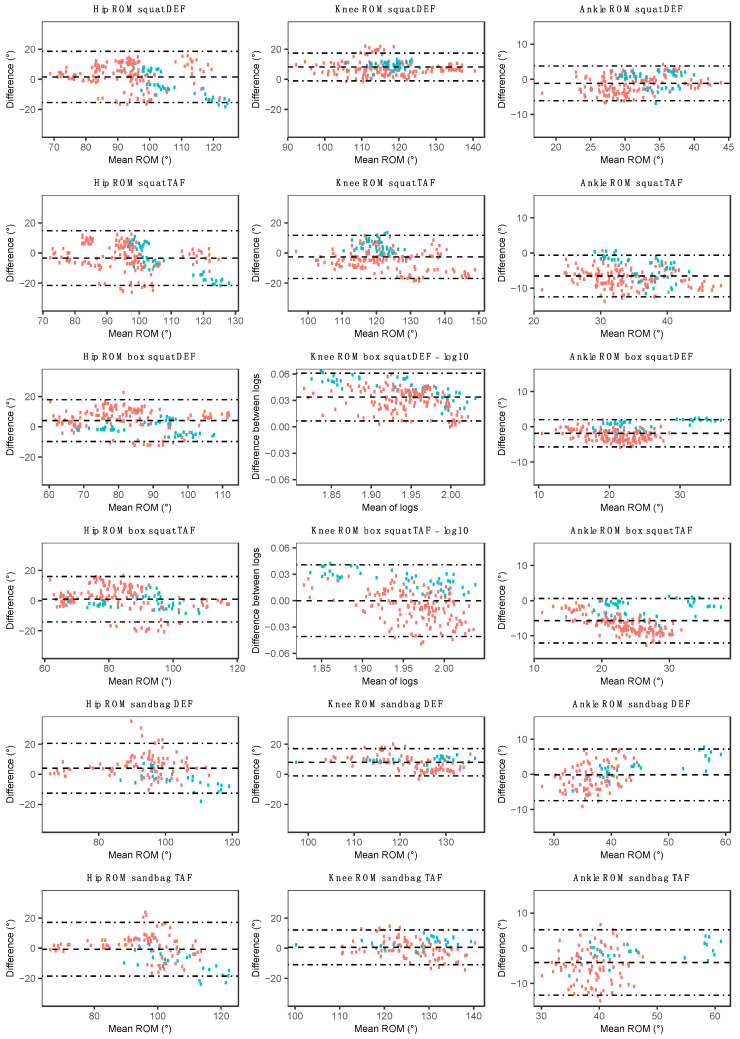
Bland–Altman plots for hip, knee, and ankle range of motion (ROM) using each MARG method (DEF/TAF) during the squat (row one/two), box squat (row three/four), and sandbag pickup (row five/six). Red data points represent male participant data, and green data points represent female participant data.

**Figure 7 sensors-20-04586-f007:**
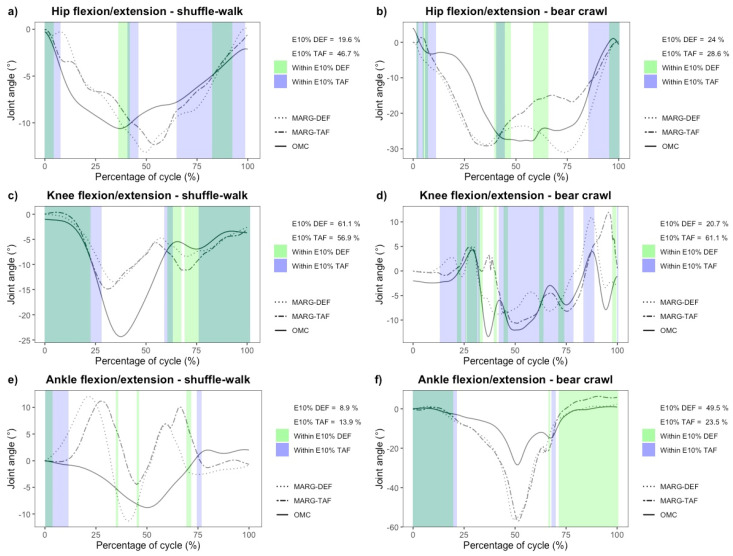
Example of preliminary time-series data and E_10%_ measurement of hip (row one), knee (row two) and ankle (row three) flexion/extension during a single stride of the shuffle-walk (column one) and bear crawl (column two).

**Figure 8 sensors-20-04586-f008:**
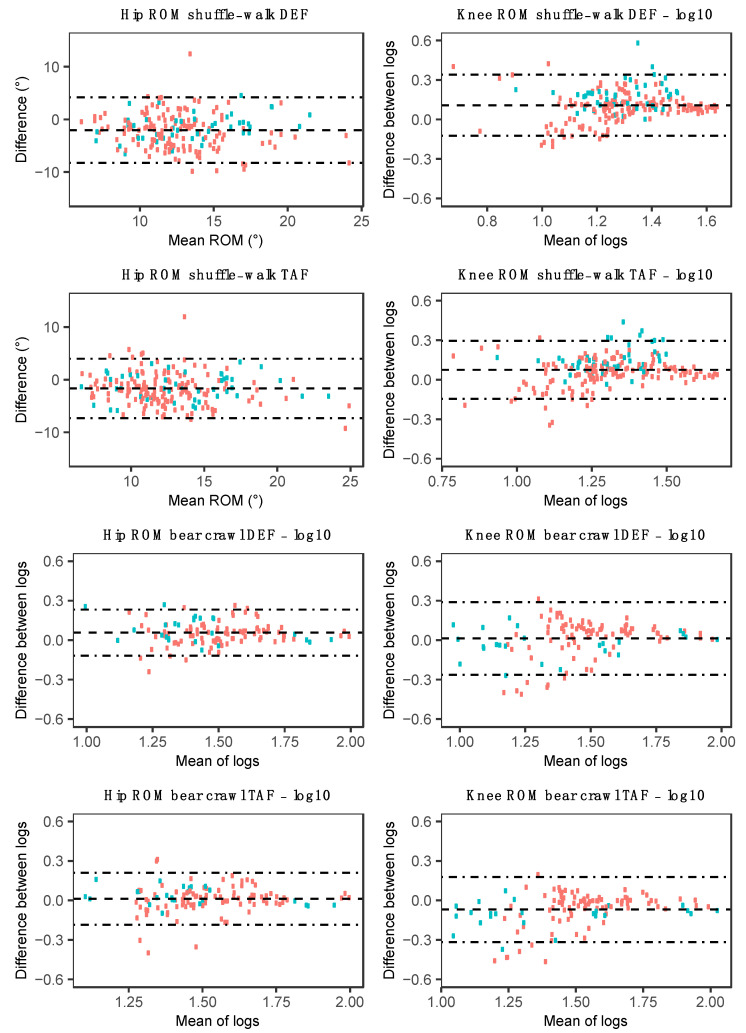
Bland–Altman plots for hip and knee ROM using each MARG method (DEF/TAF) during the shuffle-walk (row one/two) and bear crawl (row three/four). Red data points represent male participant data, and green data points represent female participant data.

**Figure 9 sensors-20-04586-f009:**
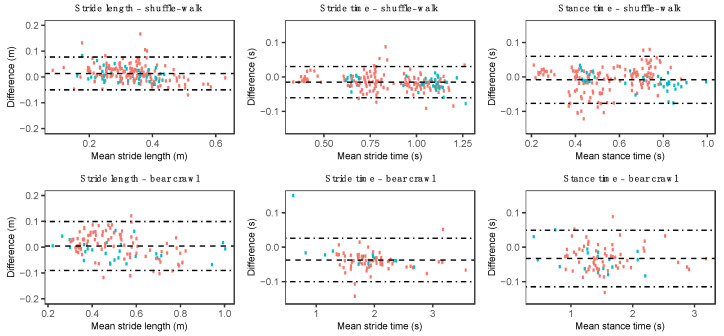
Bland–Altman plots for the stride length, stride time, and stance time during the shuffle-walk (row one) and bear crawl (row two). Red data points represent male participant data, and green data points represent female participant data.

**Table 1 sensors-20-04586-t001:** MARG device positioning.

Segment	MARG Position
Pelvis	Midway between the right and left posterior superior iliac spine
Thigh	Approximately 150 mm proximal to the lateral epicondyle of the femur
Shank	Approximately 100 mm distal to the lateral tibial condyle
Foot	Halfway between the lateral malleoli and the base of the foot

**Table 2 sensors-20-04586-t002:** MARG tuning properties.

Tuning Property	DEF Method	TAF Method
Variance of accelerometer noise (m/s^2^)^2^	1.92 × 10^−3^	3.45 × 10^−4^
Variance of gyroscope noise (rad/s)^2^	9.14 × 10^−4^	1.40 × 10^−6^
Gyroscope offset drift (rad/s)^2^	3.05 × 10^−13^	1.77 × 10^−8^
Magnetometer disturbance noise (µT^2^)	5.00 × 10^−1^	1.00 × 10^−1^
Linear acceleration compensation factor	5.00 × 10^−1^	9.00 × 10^−1^
Expected magnetic field strength (µT)	50.0	(unique to each magnetic calibration)

**Table 3 sensors-20-04586-t003:** Kinematic measures and error metrics.

	OMC	TAF MARG				DEF MARG			
	ROM (°)	ROM (°)	RMSE (°)	MAPE (%)	E_10%_ (%)	ROM (°)	RMSE (°)	MAPE (%)	E_10%_ (%)
**Hip**									
Squat	96.8 ± 11.8	100.2 ± 14.9 *	9.8	8.2 ± 6.5	95.6 ± 8.5	95.3 ± 14.3	8.8	7.6 ± 4.6	96.0 ± 7.7
Box squat	85.5 ± 12.6	84.6 ± 14.8 *	7.7	6.8 ± 6.1 *	94.4 ± 9.9 *	81.5 ± 13.7	8.1	8.0 ± 5.1	92.2 ± 13.2
Sandbag pickup	97.1 ± 11.4	97.7 ± 14.9 *	9.1	7.0 ± 5.5	87.4 ± 12.4	93.0 ± 13.0	9.3	7.0 ± 5.7	88.1 ± 13.1
Shuffle walk	12.1 ± 3.3	14.1 ± 3.7 *	3.3	25.1 ± 21.0 *	-	14.4 ± 3.7	3.8	28.6 ± 24.7	-
Bear crawl	33.3 ± 13.5	32.9 ± 12.4 *	7.1	16.5 ± 21.5	-	30.7 ± 12.2	7.7	16.7 ± 13.4	-
**Knee**									
Squat	121.2 ± 9.5	123.8 ± 11.5 *	7.7	5.1 ± 3.7 *	100.0 ± 0.4	113.0 ± 9.8	9.4	6.7 ± 3.8	100.0 ± 0.5
Box squat	91.6 ± 9.1	91.9 ± 10.8 *	4.4	4.0 ± 2.7 *	100.0 ± 0.0	84.9 ± 9.6	7.2	7.4 ± 3.0	100.0 ± 0.0
Sandbag pickup	126.8 ± 7.2	126.3 ± 8.5 *	5.9	3.7 ± 2.8 *	99.2 ± 3.3	118.8 ± 8.8	9.2	6.4 ± 3.6	98.9 ± 3.2
Shuffle walk	29.1 ± 8.9	22.9 ± 8.0 *	6.8	22.5 ± 16.5 *	-	21.1 ± 7.4	7.9	26.0 ± 13.9	-
Bear crawl	40.0 ± 20.4	44.0 ± 20.6 *	8.6	28.4 ± 40.6	-	36.7 ± 19.0	8.4	27.3 ± 30.8	-
**Ankle**									
Squat	31.2 ± 5.2	37.7 ± 5.7 *	7.2	21.9 ± 11.2 *	79.6 ± 15.3 *	32.3 ± 4.5	2.8	7.9 ± 6.1	93.9 ± 8.3
Box squat	21.1 ± 4.9	26.8 ± 5.4 *	6.6	28.6 ± 15.3 *	73.2 ± 15.0 *	23.0 ± 4.4	2.7	11.7 ± 7.5	89.5 ± 10.3
Sandbag pickup	38.9 ± 7.4	42.9 ± 6.1 *	6.2	13.9 ± 11.9 *	84.1 ± 13.4 *	39.0 ± 5.5	3.7	8.2 ± 5.7	93.8 ± 8.2

Values presented as the mean ± standard deviation where relevant. * Significant difference between the TAF and DEF method (*p* > 0.05).

**Table 4 sensors-20-04586-t004:** Bland–Altman limits of agreement.

	MARG TAF			MARG DEF		
	L−LoA	Bias	U−LoA	L−LoA	Bias	U−LoA
**Hip ROM**						
Squat (°)	−21.5	−3.4	14.8	−15.42	1.6	18.6
Box squat (°)	−14.1	1.0	16.0	−9.8	4.1	17.9
Sandbag pickup (°)	−18.5	−0.7	17.2	−12.5	4.0	20.6
Shuffle−walk (°)	−7.3	−1.6	4.0	−8.2	−2.0	4.2
Bear crawl	−0.1857 *	0.0123 *	0.2100 *	−0.1182 *	0.0572 *	0.2325 *
**Knee ROM**						
Squat (°)	−16.9	−2.6	11.7	−1.1	8.2	17.4
Box squat	−0.0409 *	−0.0001 *	0.0407 *	0.0065 *	0.0337 *	0.0608 *
Sandbag pickup (°)	−11.0	0.6	12.1	−1.1	8.0	17.1
Shuffle−walk	−0.1463 *	0.0746 *	0.2954 *	−0.1239 *	0.1079 *	0.3398 *
Bear crawl	−0.3165 *	−0.0692 *	0.1781 *	−0.2633 *	0.0129 *	0.2891 *
**Ankle ROM**						
Squat (°)	−12.5	−6.5	−0.6	−6.1	−1.2	3.8
Box squat (°)	−12.0	−5.7	0.6	−5.8	−1.9	2.0
Sandbag pickup (°)	−13.4	−4.1	5.2	−7.5	−0.1	7.2
**Spatiotemporal**						
Stride length (m)	−0.050	0.013	0.077	−0.090	0.004	0.099
Stride time (s)	−0.061	−0.015	0.030	−0.100	−0.037	0.0256
Stance time (s)	−0.077	−0.008	0.060	−0.115	−0.033	0.049

Positive bias represents underestimation by the MARG method and negative bias represents overestimation by the MARG method; * log transformed data (unitless); L-LoA, lower limits of agreement; U-LoA, upper limits of agreement.

**Table 5 sensors-20-04586-t005:** Spatiotemporal measures of the shuffle-walk and bear crawl.

	OMC			MARG		
	Stride Length (m)	Stride Time (s)	Stance Time (s)	Stride Length (m)	Stride Time (s)	Stance Time (s)
Shuffle-walk	0.339 ± 0.086	0.846 ± 0.219	0.568 ± 0.179	0.326 ± 0.096	0.861 ± 0.224	0.577 ± 0.177
Bear crawl	0.515 ± 0.157	1.912 ± 0.479	1.502 ± 0.497	0.511 ± 0.175	1.949 ± 0.489	1.535 ± 0.500

Values presented as the mean ± standard deviation.

**Table 6 sensors-20-04586-t006:** Error metrics of spatiotemporal measures.

	Stride Length		Stride Time		Stance Time	
	RMSE (m)	MAPE (%)	RMSE (s)	MAPE (%)	RMSE (s)	MAPE (%)
Shuffle-walk	0.035	8.2 ± 7.9	0.028	2.6 ± 2.1	0.036	5.2 ± 5.9
Bear crawl	0.048	7.8 ± 5.7	0.049	2.4 ± 2.5	0.053	3.2 ± 2.8

Values presented as the mean ± standard deviation where relevant.

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
