# Peer review of "Validation of Spatiotemporal and Kinematic Measures in Functional Exercises Using a Minimal Modeling Inertial Sensor Methodology"

_sensors, 2020, doi:10.3390/s20164586_

Round 1

Reviewer 1 Report

This paper presents an inertial-based sensor approach for measuring spatiotemporal and kinematic parameters of the lower-limb joints while performing exercise. The approach relies on the MARG methodology comprising four inertial sensors, each encompassing triaxial accelerometers, gyroscopes, and magnetometers. MARG measurements were compared to the measurements obtained by video captured on-body markers. Agreement between both approaches spans all ranges: from acceptable to unacceptable. Results are well discussed.

Overall, the paper addresses an interesting topic on motion sensors. The paper is well written; it is easy to read, and to follow. All concepts are comprehensively described. The proposed methodology is appropriate and results are thoughtfully discussed. I consider the manuscript to be a solid contribution to Sensors and I recommend its acceptance with minor revisions. In particular:

A) Content

- The state of the art review on motion capture systems is sparse. Only IMU and video marker technologies are mentioned and very briefly.

- Plots in figs. 7 to 9: it would be interesting to identify with a different color the data belonging to the three female participants. Being their anthropometrics significantly different to their male counterparts, it would be useful to locate their performance in the plots.

B) Format:

- Plots in figure 9 are not individually identified (a, b, c,…). Later, lines 300, 302, 309, and 310 make reference to figures 9a, 9b, etc.

Reviewer 2 Report

The  paper is well organized 

 The fusion algorithms are very briefly described.

Reviewer 3 Report

This paper describes a methodology for evaluating physical exercises using spatiotemporal and kinematic measures. The paper is well presented, and the results are clearly showed. From my point of view, I think the paper contribution is small. There is not any algorithm or system proposal, only a methodology based on well-known algorithms and sensors. But the analysis carried out is complete and the conclusions are supported by a detailed analysis of several measurements.

General comments

  • I think is necessary to expand the state-of-the-art analysis. In this paper, only lines 55-60 describe previous methodologies. This description should be extended with more references and more detailed analysis of this previous work.
  • In figure 4, I’d suggest including more information. It seems that the difference between both methods is only a low-pass filter but then in the text, also the initial settings are different. I think it is interesting to describe more in detail these settings and the differences.
  • Have you analyzed the error due to transformation estimations? Perhaps part of the error comes from these mathematical transformations.
  • In Figure 6 e) there are important difference, not only in the Joint Angle but also in the time evolution (curve shape). The curve is very different. Some explanations about these differences would be appreciate.
  • Section 3.1 includes many figures and tables but the explanation is very small. I think it is necessary to comment/describe all the presented results. There are important differences between physical exercises. The characteristics of each exercise have correlation with the differences found in the analysis. I’d suggest expanding this study. Some explanations are included in the discussion section, but I think an expanded description of the results would be appreciated.
  • The authors claim the need of a further development of the proposed method. I’d suggest including a “future work” section describing some proposals to deal with the limitations analyzed and described in the paper.

Minor comments

  • In any section, before including a subsection, it is recommendable to include an introductory paragraph.
  • There are many initials. I’d suggest including a glossary.

Round 2

Reviewer 3 Report

I think the authors have addressed all my comments and the paper can be accepted